# Variation in the frozen lesion size according to the non-occluded application duration and technique for cryoballoon ablation

Tetsuma Kawaji [1,2]*, Bingyuan Bao[1], Shun Hojo[1], Yuji Tezuka[1], Kenji Nakatsuma[1], Shintaro Matsuda[1], Masashi Kato[1], Takafumi Yokomatsu [1], Shinji Miki[1]

**1** Department of Cardiology, Mitsubishi Kyoto Hospital, Kyoto, Japan, **2** Department of Cardiovascular Medicine, Graduate School of Medicine, Kyoto University, Kyoto, Japan

* kawaji@kuhp.kyoto-u.ac.jp

**Data Availability Statement:** All relevant data are within the paper.

**Funding:** The author(s) received no specific funding for this work.

## Abstract

### Objective

The frozen lesion formation created by cryoballoon ablation, especially with non-occluded applications, has not been fully evaluated. This study aimed to validate the lesion size under different cryoballoon ablation settings: application duration, push-up technique, and laminar flow.

### Methods

The frozen lesion size was evaluated immediately after ending the freezing with three different application durations (120, 150, and 180 seconds) in porcine hearts (N = 24). During the application, the push-up technique was applied at 10, 20, and 30 seconds after starting the freezing with or without laminar flow.

### Results

The lesion size was significantly correlated with the nadir balloon temperature (P<0.001). The lesion volume became significantly larger after 150 seconds than 120 seconds (1272mm$^3$ versus 1709mm$^3$, P = 0.004), but not after 150 seconds (versus 1876mm$^3$ at 180 seconds, P = 0.29) with a comparable nadir balloon temperature. Furthermore, the lesion volume became significantly larger with the push-up technique with the largest lesion size with a 20-second push-up after the freezing (1193mm$^3$ without the push-up technique versus 1585mm$^3$ with a push-up at 10 seconds versus 1808mm$^3$ with a push-up at 20 seconds versus 1714mm$^3$ with a push-up at 30 seconds, P = 0.04). Further, the absence of laminar flow was not associated with larger lesion size despite a significantly lower nadir balloon temperature.

### Conclusion

The frozen lesion size created by cryoballoon ablation became larger with longer applications at least 150 seconds and with a push-up technique especially at 20 seconds after the freezing.

**Competing interests:** The authors declared that no competing interests exist.

## Introduction

Cryoballoon ablation delivers cryoenergy to the antrum of the pulmonary vein (PV) with an occlusion of the PV during the PV isolation. The utility of cryoballoon ablation for paroxysmal atrial fibrillation (AF) in terms of the safety, efficacy, and simplicity has been documented [1]. Recent randomized clinical trials have established its role as a first-line therapy for paroxysmal AF [2, 3]. Furthermore, the cryoballoon is utilized for non-occluded applications such as the posterior wall isolation, exhibiting a high success rate in the complete block of the left atrial roof line [4–6]. While the clinical utility of cryoballoon ablation has been proven in several studies, the lesion created by cryoballoon ablation, especially for non-occluded applications, remains inadequately assessed. We hypothesize that factors like application duration and various techniques such as push-up (raise-up) technique and rapid ventricular pacing during the freezing largely influence the lesion size created by cryoballoon ablation. The aim of this study was to validate the frozen lesion sizes under varied settings (application duration, push-up technique, and laminar flow) for non-occluded cryoballoon applications.

## Methods

### Experimental design

We examined the size of the frozen lesion immediately after the end of freezing using various cryoballoon settings (application duration, push-up technique, and laminar flow). All methods were carried out in accordance with the relevant guidelines and regulations. We prepared a commertially available porcine hearts (ventricles), which were mounted on a platform and placed in a lucid chamber filled with circulating normal saline maintained at 37°C with a general impedance of 100 Ω to mimic an intracardiac blood flow (Fig 1A). The laminar flow was generated by a peristaltic pump. We delivered cryoballoon (Arctic Front Advance 28mm; Medtronic., Dublin, Ireland) applications from the epicardial surface with three different application durations (120, 150, and 180 seconds) in porcine hearts. During the freezing, we brought the northern hemisphere of the balloon in contact with the epicardial surface, and the contact force was maintained to fixate the balloon's position. Furthermore, a push-up technique was attempted at different timings during the freezing (Fig 1B). In the push-up technique, we pushed the balloon straight onto the surface without any rotation until the balloon became almost sideways, and the resistance to the pushing did not increase significantly. Finally, we delivered 24 cryoballoon applications for 120, 150, and 180 seconds with and without the push-up technique at 10, 20, and 30 seconds after the start of the freezing under the presence or absence of laminar flow. During the freezing using the push-up technique, as well as without the push-up technique, the contact force was maintained to fixate the balloon position. During each cryoballoon application, we continuously recorded the balloon temperature and ice cap formation.

### Measurements of lesion size

We examined the frozen lesion sizes in the porcine hearts immediately after the end of the freezing. Each ablation lesion was measured in the cross-section through its center (major and minor axes) on the assumption that the ablation lesion was an ellipse shape (Fig 1C). The surface length ($Ws_1$) in the major axis (front-rear axis), surface width ($Ws_2$) in the minor axis (left-right axis), and lesion depth (Dt) were measured. The surface area and lesion volume were calculated as below:

$$\text{Surface area} = \pi(Ws_1/2)(Ws_2/2)$$

$$\text{Lesion volume} = 2/3 \times \pi(Ws_1/2)(Ws_2/2)Dt$$

# Experimental study

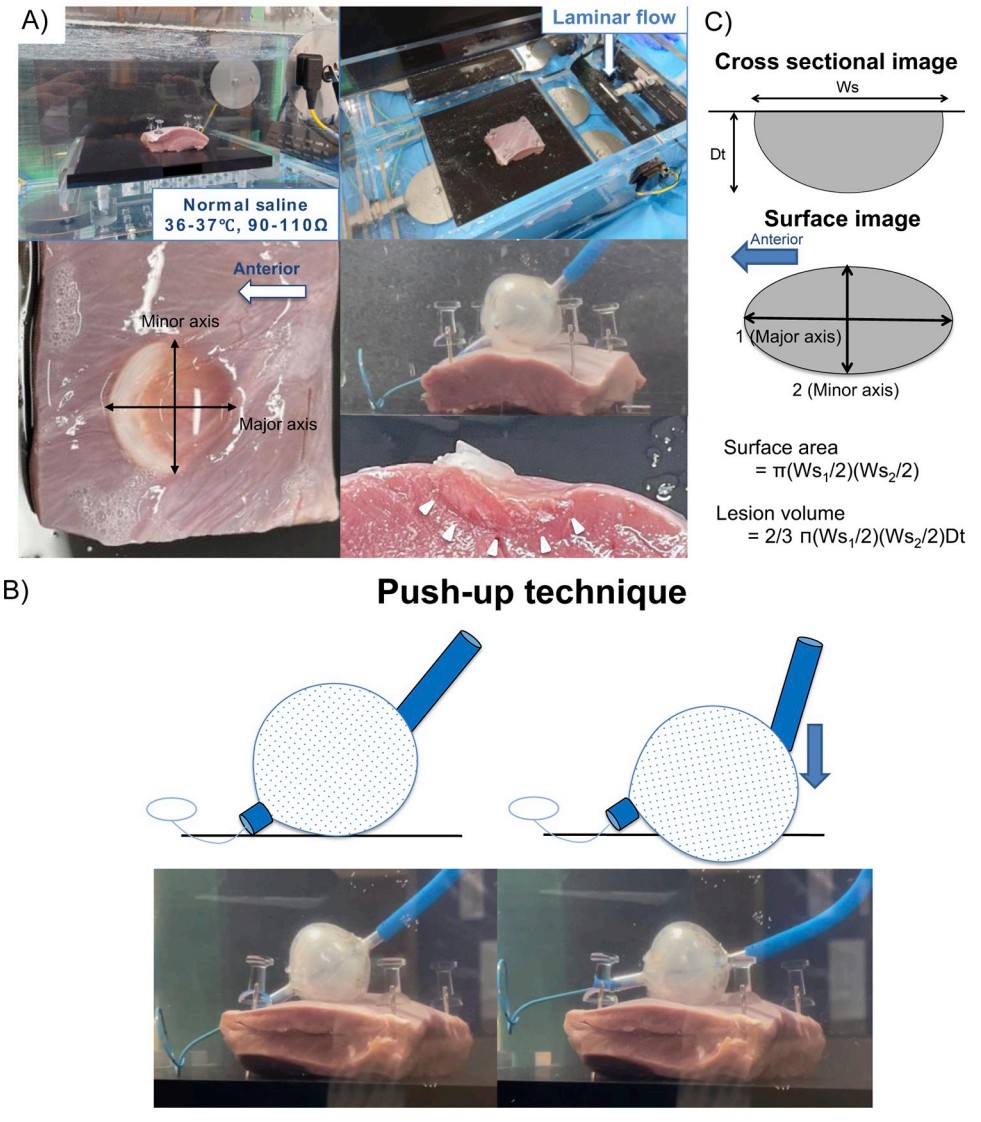

**Fig 1. Schematic representation of this experimental study.** A) Ex-vivo setup, B) push-up technique, and C) lesion measurements.

## Outcome measurements and statistical analysis

The outcome measurements in this study were the nadir balloon temperature, thawing time, time required for the ice cap formation covering the front of the balloon, and balloon temperature at the time of the push-up technique for the freezing data, and the lesion length, width, depth, surface area, and lesion volume for the lesion size. The correlation between the nadir balloon temperature and each measured lesion size was assessed by Pearson's correlation coefficient. Each measured lesion value was presented as the median with the interquartile range, and compared between the various settings using the Wilcoxon rank sum test. Statistical analyses were performed using JMP 10 (SAS Institute Inc, Cary, NC) software. Probability values of $p < 0.05$ was considered statistically significant.

**Table 1. Study results.**

| | Overall<br>N = 24 |
|---|---|
| Balloon temperature at 30 seconds (˚C) | -36 (-38 - -34) |
| Balloon temperature at 60 seconds (˚C) | -42 (-44 - -40) |
| Nadir balloon temperature (˚C) | -45 (-48 - -45) |
| Thawing time (seconds) | 37 (28–41) |
| Time required for ice formation (seconds) | 33 (31–37) |
| *Lesion size characteristics* | |
| Lesion length (mm) | 20.5 (19.0–21.5) |
| Lesion width (mm) | 21.0 (19.6–22.5) |
| Lesion depth (mm) | 6.9 (6.0–7.5) |
| Surface area (mm$^2$) | 329.9 (300.4–373.6) |
| Lesion volume (mm$^3$) | 1564 (1250–1826) |

Continuous variables are presented as the median with the interquartile range.

## Results

### Lesion characteristics

The cryoballoon applications were applied to 24 sites. The nadir balloon temperature was -45 (-48 - -45)˚C and thawing time 37 (28–41) seconds (Table 1). An ice cap formation was seen at 33 (31–37) seconds. The representative cryoballlon temperature during the freezing is shown in Supplemental Fig 1 in S1 File.

Most frozen lesions created by the cryoballoon applications gradually diminished within a few minutes after the end of the freezing. The surface lesion was almost a perfect circle with an ice cap formation for the anterior lesions. All lesion sizes (lesion length, width, depth, surface area, and lesion volume) were inversely correlated with the nadir balloon temperature (R = 0.40, 0.50, 0.70, 0.49, and 0.65, all P<0.001) (Fig 2).

### Relationships between the lesion size and cryoballoon settings

Regarding the lesion size according to the cryoballoon application duration, the freezing data including the nadir balloon temperature was almost comparable between 120 seconds, 150 seconds, and 180 seconds (Fig 3). Although the lesion volume was significantly larger for 150 seconds than 120 seconds (1272 mm$^3$ versus 1709 mm$^3$, P = 0.04), the lesion volume was comparable between 150 seconds and 180 seconds (versus 1876 mm$^3$, P = 0.29).

Regarding the lesion size according to the use of the push-up technique, the lesion size was significantly larger for applications with the push-up technique than for those without (lesion volume: 1709 mm$^3$ versus 1193 mm$^3$, P = 0.007) (Supplemental Fig 2 in S1 File). Although the balloon temperature at the time of the push-up significantly differed among the different push-up timings (-6.5˚C at 10 seconds versus -27˚C at 20 seconds versus -37˚C at 30 seconds, P<0.001) (Supplemental Fig 3 in S1 File), the final freezing data including the nadir balloon temperature were almost comparable (Fig 4). The lesion size with the push-up technique applied at 20 seconds after the freezing was the largest (lesion volume: 1585 mm$^3$ with a push-up at 10 seconds versus 1808 mm$^3$ with a push-up at 20 seconds versus 1714 mm$^3$ with a push-up at 30 seconds, P = 0.04).

Regarding the freezing data according to the laminar flow, the nadir balloon temperature was significantly lower with the absence of laminar flow than with the presence of it (-48˚C

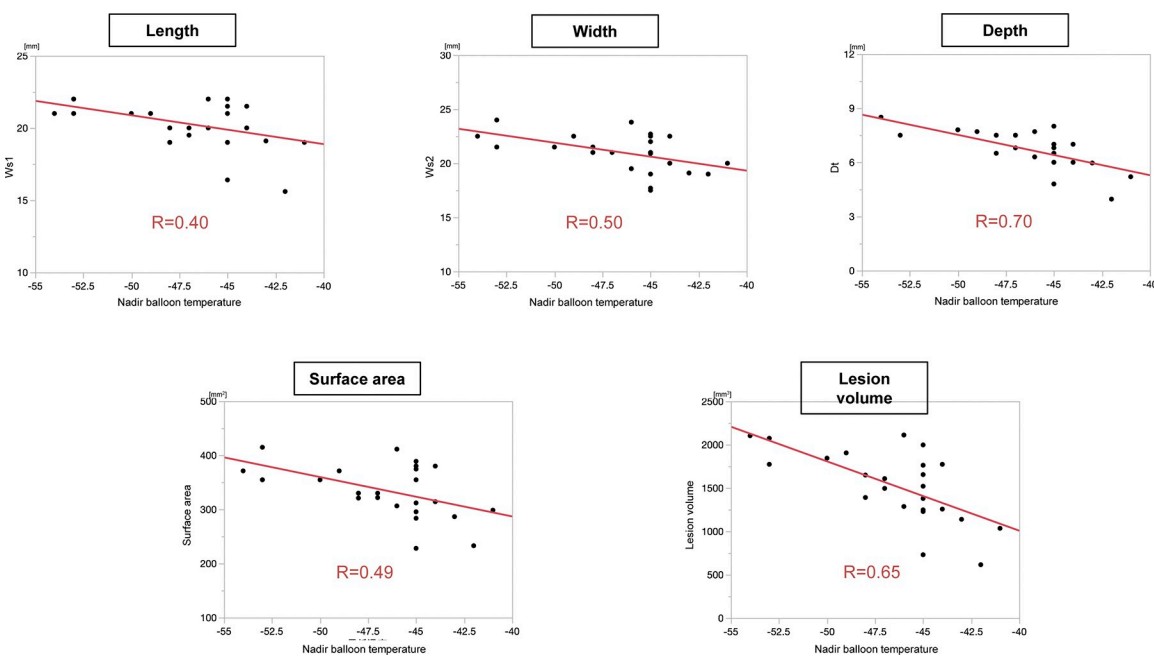

**Fig 2. Correlations between the lesion size and nadir balloon temperature.**

versus -45˚C, P<0.001). Moreover, the time required for an ice cap formation covering the front of the balloon was significantly shorter (32 seconds versus 36 seconds, P = 0.03) and thawing time significantly longer (40 seconds versus 20 seconds, P<0.001) (Fig 5). However, the lesion size was not significantly different between the absence and the presence of laminar flow.

## Discussion

In this experimental study, we measured the frozen lesion size after non-occluded cryoballoon applications using various settings and found the following: (a) The lesion size was significantly correlated with the nadir balloon temperature. (b) The lesion size became larger with longer applications of less than 150 seconds, but not with those longer than 150 seconds. (c) The lesion size became significantly larger with the push-up technique, especially at 20 seconds after the freezing. (d) The absence of laminar flow was not associated with larger lesion size despite a significantly lower nadir balloon temperature.

Cryoenergy induces cell necrosis through extra- and intracellular ice formation during freezing, which subsequently melts during rapid thawing and results in a delayed cellular damage [7]. Histopathological lisions created by occluded applications of the cryoballoon have been confirmed in animal studies, and are characterized as transmural circumferential lesions in the antrum and and PVs [8–12]. Additionally, Khairy et al. demonstrated homogenous lesions with smooth and sharp demarcations from the intact myocardium created by cryoenergy in contrast to less well-circumscribed lesions with serrated edges created by radiofrequency energy, which resulted in thrombus adhesions [9]. Moreover, Hirao et al. reported an autopsy case at 6 months after a cryoballoon ablation [13]. The lesions created by PV isolation of the cryoballoon were gross-pathologically unclear, but microscopic-pathologically replaced by enough fibrous tissue to form a transmural lesion. In contrast, the lesions at the cavotricuspid isthmus created by a radiofrequency catheter were characterized by thin and shallow fibrous tissue with subintimal hemorrhaging.

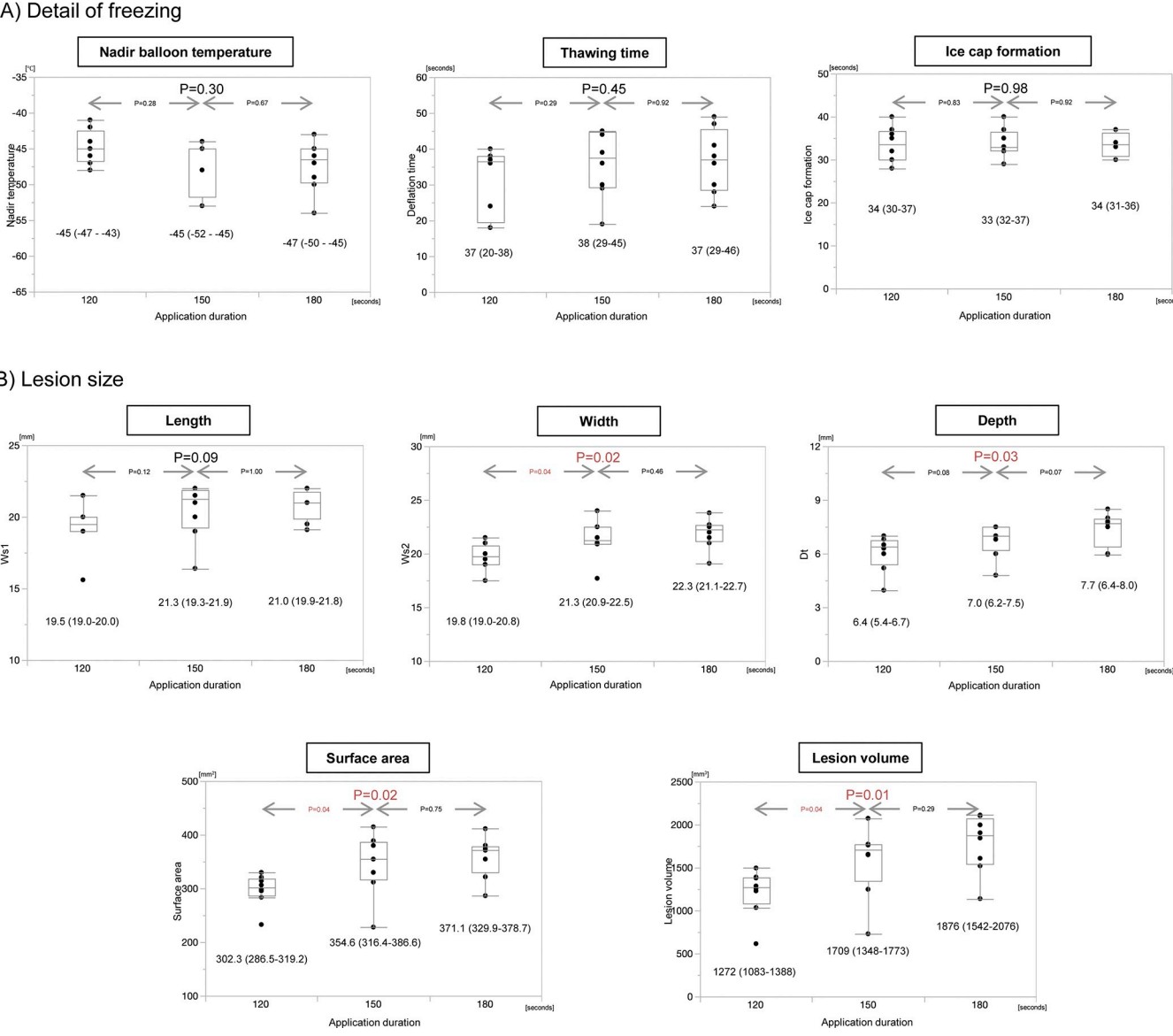

**Fig 3. Study results according to the application duration.** A) Details of the freezing and B) lesion size. * statistically significant, ** marginally significant.

Although only few studies has assessed the lesions created by cryoenergy as described above, the clinical utility of the cryoballoon using occluded applications for the PV isolation has been reported in several randomized clinical trials [1–3]. Recently, the clinical utility of the cryoballoon using non-occluded applications such as posterior wall isolation for persistent AF has been reported. Notably, the acute success rate in achieving a complete block of the left atrial roof line by cryoballoon ablation is remarkable compared with radiofrequency catheter ablation [4–6]. The reason for the high success rate is attributed to the broad and deep lesions created by the cryoenergy applications, reaching the epicardial connections such as the septo-pulmonary bundle [14], which often causes difficulty in creating a complete block of the left atrial roof line with radiofrequency catheter ablation. Our study findings revealed that long application durations (at least 150 seconds) and a push-up technique (especially at 20 seconds after the freezing) for non-occluded applications resulted in broad and deep frozen lesions.

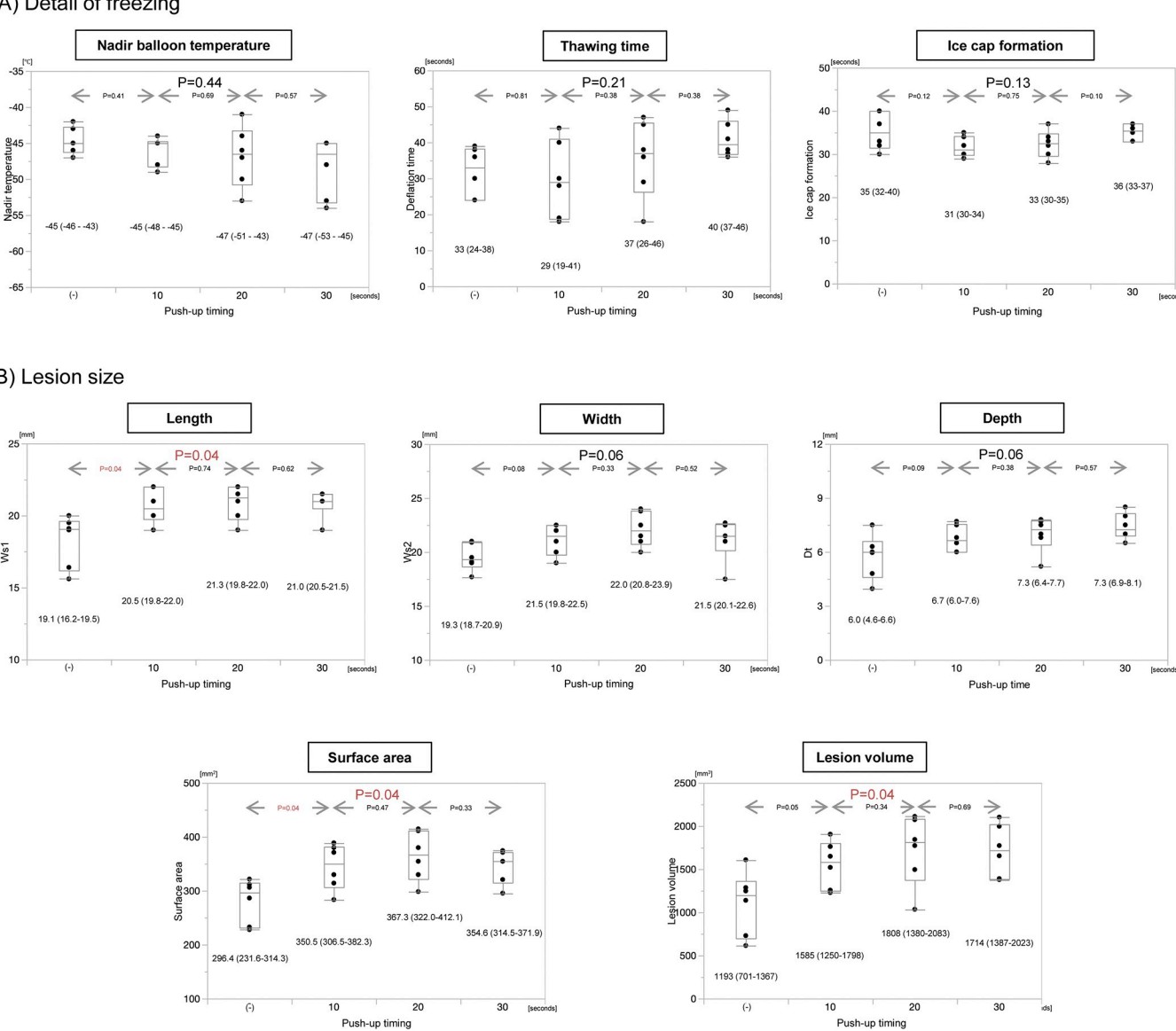

**Fig 4. Study results according to the push-up technique.** A) Details of the freezing and B) lesion size. * statistically significant, ** marginally significant.

Dubuc et al. also previously reported that the lesion size evaluated by real-time intracardiac echocardiography increased within the first 3 minutes of freezing the atrial myocardium but not after 3 minutes [15]. Lichter et al. also reported the increase of freeze-zone assessed by real-time MRI according to application duration was seen within 120 seconds, but not beyond 120 seconds [16]. Consistent with these studies, our study demonstrated that the intramural frozen lesion size grew with longer application durations up to 150 seconds, but not beyond 150 seconds of the freezing, which might indicate that 150 seconds was an optimal freezing duration for non-occluded applications. A previous histopathological assessment o in animal models showed a comparable lesion length and depth under various application durations (2 minutes versus 4 minutes [12] and 3 minutes versus 4 minutes) [11], but the neointimal thickness, which indicates the degree of myocardial injury, within 4 minutes of the cryballoon application was significantly larger than within 2 minutes. These discrepancy might be attributed to

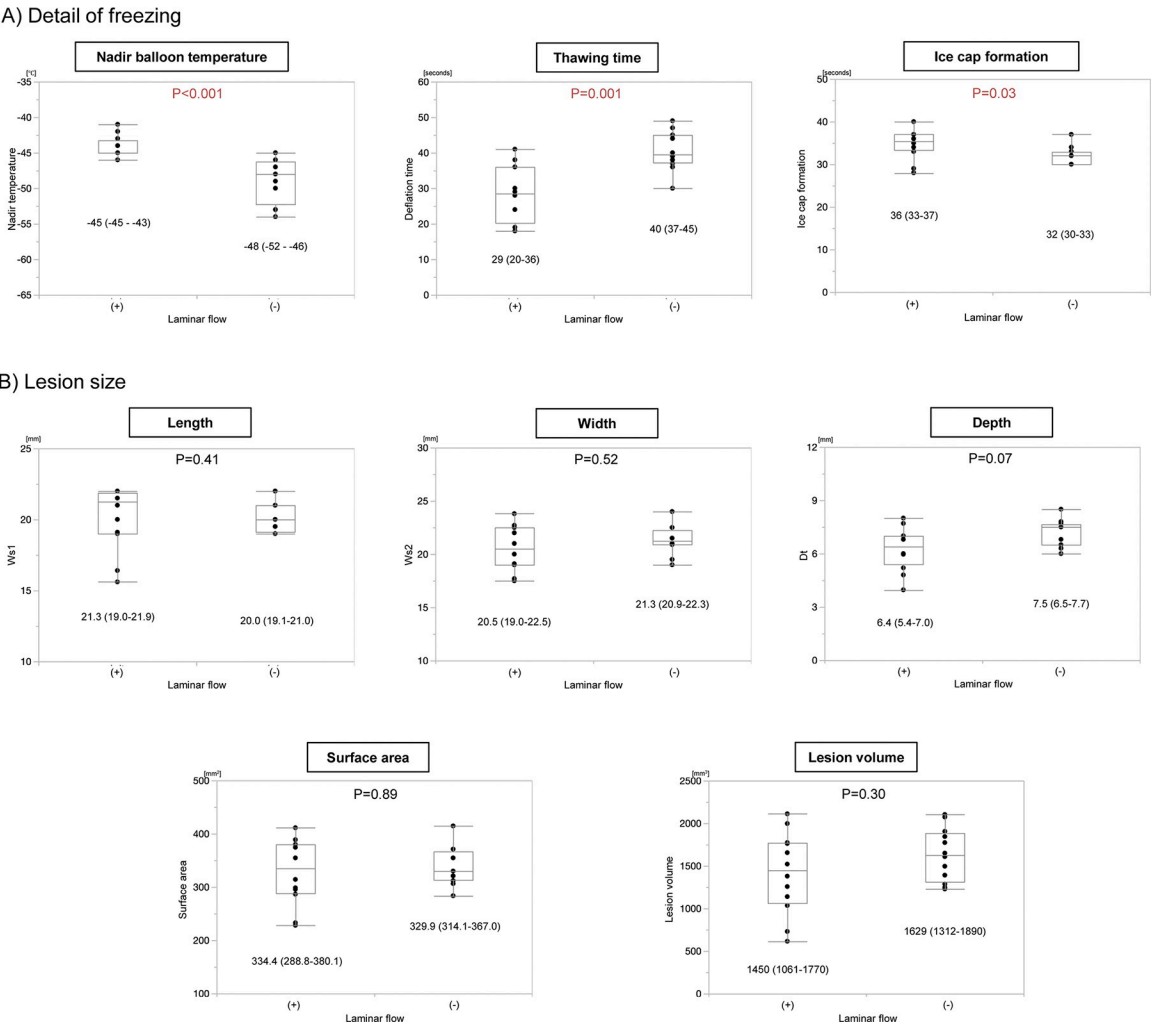

**Fig 5. Study results according to the laminar flow.** A) Details of the freezing and B) lesion size. * statistically significant, ** marginally significant.

the difference among the electrically dormant, non-viable, edematic and frozen (partially reversible) lesions [17]. Determining the optimal application duration of cryoballoon ablation, especially during non-occluded applications, remains elusive.

Our study firstly assessed the impact of the push-up technique and the absence of laminar flow, which assumed an rapid ventricular pacing technique during a non-occluded cryoballoon application on the lesion size. The push-up technique increased the lesion length, width, and depth, potentially due to the heightened contact pressure and the expanded contact area to the equatorial plane of the balloon. The push-up technique, especially at 20 seconds after the freezing, is beneficial in creating larger frozen lesion, attributing complete block of roof line by non-occluded cryoballoon applications in daily clinical practice use.

In this study, despite achieving effective freezing, the absence of laminar flow, assuming rapid ventricular pacing during non-occluded cryoballoon application, did not notably increase lesion size, except for a non-significant increase in lesion depth. This discrepancy between freezing efficacy and lesion size primarily stemmed from the placement of the temperature sensor on the proximal side of the balloon, causing the recorded balloon temperature to predominantly reflect blood temperature rather than the actual tissue temperature. This

discrepancy between freezing effectiveness and lesion size was also evident in the assessment of the impact of application duration and the push-up technique. In clinical practice use, Nishimura et al. performed a randomized clinical trial to assess the utility of rapid ventricular pacing on acute success rate in achieving a complete block of the left atrial roof line by cryoballoon applications [18]. Although the use of rapid ventricular pacing notably decreased the nadir balloon temperature, it did not affect the acute success rates in patients who received a 180-second cryoballoon application, but increased the acute completion rates in patients who experienced early interruption of the cryoballoon application due to prominent decline of esophageal temperature. Considering these insights, despite the absence of laminar flow resulting in non-significant increase in lesion depth, a decrease of blood temperature around the balloon by effective freezing may indirectly enhance freezing efficiency during non-occluded application.

## Study limitations

Several limitations of this experimental study should be noted. First, this experimental study was conducted on the epicardial side of isolated porcine ventricular tissue, so there was no effect of the respiratory movement or cardiac beating. On the other hand, in daily clinical practice, the cryoballoon is used on the endocardial side of the atrial wall, which is thinner and softer than the ventricular wall, and with non-occluded application for atrial roof area. The frozen target and situation such as stability and direction are largely different from the non-occluded cryoballoon application in daily clinical practice use. Values such as the freezing duration and/or balloon temperature cannot be directly extrapolated to clinical use on beating human atrium. Second, we measured only the visually frozen lesions including reversible lesion, but not the histopathologically nor electrically degenerated lesion size, which is generally considered to be smaller than the frozen lesion size. We need further study to assess the histopathological lesion size created by non-occluded cryoballoon applications. However, the sizes of those lesions are correlated with each other. The impact of the technique on frozen lesion size in this study may be consistent on these lesion sizes. Third, the absence of the laminar flow in this study may not adequately simulate the rapid ventricular pacing on the beating heart in daily clinical practice. Fourth, contact force of the cryoballoon in this study was not controlled. The difference of contact force during the freezing in various settings may influence the study results. Finally, the relatively small sample size in this study precluded drawing any definitive conclusion.

## Conclusion

The frozen lesion size created by cryoballoon ablation became larger with longer applications at least 150 seconds and with the push-up technique especially at 20 seconds after the freezing, but not with the absence of laminar flow. These techniques are important to create the large frozen lesion during non-occluded cryoballoon applications.

## Supporting information

**S1 File. Supplemental materials.**
(DOCX)

## Acknowledgments

The authors thank all the members of the experimental study for their contribution to this study.

## Author Contributions

**Data curation:** Tetsuma Kawaji.

**Formal analysis:** Tetsuma Kawaji.

**Investigation:** Tetsuma Kawaji, Bingyuan Bao, Shun Hojo, Yuji Tezuka, Kenji Nakatsuma, Shintaro Matsuda, Masashi Kato, Takafumi Yokomatsu, Shinji Miki.

**Methodology:** Tetsuma Kawaji.

**Writing – original draft:** Tetsuma Kawaji.

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
