## [Decision Letter · Decision Letter 0]

12 Dec 2023

PONE-D-23-36076Variation in the frozen lesion size according to the non-occluded application duration and technique for cryoballoon ablationPLOS ONE

Dear Dr. Kawaji,

Thank you for submitting your manuscript to PLOS ONE. After careful consideration, we feel that it has merit but does not fully meet PLOS ONE’s publication criteria as it currently stands. Therefore, we invite you to submit a revised version of the manuscript that addresses the points raised during the review process.

We look forward to receiving your revised manuscript.

Kind regards,

Gaetano Santulli, MD, PHD, FAHA

Academic Editor

PLOS ONE

Journal Requirements:

2. Thank you for submitting the above manuscript to PLOS ONE. During our internal evaluation of the manuscript, we found significant text overlap between your submission and previous work in the [introduction, conclusion, etc.].

Please revise the manuscript to rephrase the duplicated text, cite your sources, and provide details as to how the current manuscript advances on previous work. Please note that further consideration is dependent on the submission of a manuscript that addresses these concerns about the overlap in text with published work.

[If the overlap is with the authors’ own works: Moreover, upon submission, authors must confirm that the manuscript, or any related manuscript, is not currently under consideration or accepted elsewhere. If related work has been submitted to PLOS ONE or elsewhere, authors must include a copy with the submitted article. Reviewers will be asked to comment on the overlap between related submissions (http://journals.plos.org/plosone/s/submission-guidelines#loc-related-manuscripts).]

We will carefully review your manuscript upon resubmission and further consideration of the manuscript is dependent on the text overlap being addressed in full. Please ensure that your revision is thorough as failure to address the concerns to our satisfaction may result in your submission not being considered further.

Reviewers' comments:

Reviewer's Responses to Questions

**Comments to the Author**

1. Is the manuscript technically sound, and do the data support the conclusions?

Reviewer #1: Partly

Reviewer #2: No

Reviewer #3: Yes

Reviewer #4: Yes

2. Has the statistical analysis been performed appropriately and rigorously? 

Reviewer #1: No

Reviewer #2: Yes

Reviewer #3: Yes

Reviewer #4: Yes

3. Have the authors made all data underlying the findings in their manuscript fully available?

Reviewer #1: Yes

Reviewer #2: Yes

Reviewer #3: Yes

Reviewer #4: Yes

4. Is the manuscript presented in an intelligible fashion and written in standard English?

Reviewer #1: Yes

Reviewer #2: No

Reviewer #3: Yes

Reviewer #4: Yes

5. Review Comments to the Author

Reviewer #1: The authors wrote an interesting manuscript entitled: "Variation in the frozen lesion size according to the non-occluded application duration and technique for cryoballoon ablation". This is an interesting ex vivo study that investigates the lesion size in myocardium using a cryogenic balloon with different settings.

Overall, the study appears technically sound, but does have significant limitations that the authors also addressed; these are: 1) Epicardial application of cryoenergy in ventricular myocardium. 2) Application in a non-beating model without any respiratory motion.

The scientific questions that the authors ask are all related to cryoablation in atrial myocardium, it is unclear to me why no atrial myocardium was used in this model. This would also have allowed to conduct this study from the endocardium and at the locations that are of interest for each technique; i.e. posterior wall, roof, PVs. I would encourage the authors to perform this study actually from the endocardium of atrial myocardium. They should provide us with comparison data regarding the biophysics of cryoballoon ablation in the latter.

I am not sure that the push up technique can be simulated in an ex vivo model because there are so many variables that impact the balloon orientation, most importantly probably position and orientation of the sheath. Therefore, these data are not easily translatable to the clinical arena.

The p value <0.05 should be used as only cut-off.

I would also have the language in the introduction and discussion professionally reviewed.

Reviewer #2: Kawaji et al report on cryoballoon ablation lesion size for non-occlusive (i.e. non-PV isolation) using an ex vivo model. The authors characterize the lesion size at various application durations, using different balloon positioning techniques, and with and without laminar flow. I have the following comments:

Cryoballoon ablation is clinically used in a beating heart to treat atrial arrhythmias. This study uses an ex vivo model with ventricular tissue. The fundamental limitation of this study is the model and lack of prior validation. Cryoablation is performed in a complex anatomic and thermal environment and it is unclear whether findings from this model have clinical relevance. Furthermore, as mentioned by the authors, cryoablation is performed to chronically interrupt electrical conduction in the heart and is not certain to what degree acute lesion size, as assessed with this investigation, correlates with the intended electrical effects of ablation.

It is a bit difficult to contextualize the volume / lesion size findings in this study. When performing ablation in the atrium, the goal is for transmural lesions and a complete line of block. Lesions that are too deep/large run the risk of collateral damage to extracardiac structures/conduction system or inadvertent electrical isolation of the LAA.

There is frequent reference to the "overdrive ventricular pacing technique" but is not clear how this is any relevance to the current model, which is performed in non-beating tissue.

In the Discussion, the authors state "The push-up technique, especially at 20 seconds after freezing, was useful for creating a complete block line with nonoccluded cryoballoon applications". Unless I am missing something, there was no assessment for electrical block in this investigation; such a statement over interprets the findings.

Reviewer #3: Interesting article that has the potential to contribute greatly to human cases of cryoablation. The cryoablation technique is relatively new and needs to be improved, and basic research helps immensely with this.

Reviewer #4: The authors are attempting to create an experimental model of non-occlusive cryoablation technique using a cryoballoon and define optimal “in-vitro” ablation parameters.

It appears that optimal application duration of lesions is 150 sec with apposition against the tissue.

Methodology

Authors are attempting to approximate the in-vivo ablation environment.

Quite appropriately authors acknowledge limitations of this environment, I have several additional concerns:

1. Ventricular myocardium certainly creates a different “cold sink” than atrial myocardium.

2. Lack of tissue perfusion has potential for affecting lesion formation.

3. There is no quantification of force applied hence potential for error.

Discussion

I am surprised by the findings of no considerable difference in lesion size with or without laminar flow of the medium surrounding the balloon – authors seem not to have a good sense of what the mechanism may be. I would postulate as well to remove any implication in the discussion that these lesions are larger since they are not statistically different. Therefore, the findings here are not consistent with clinical findings quoted in reference 18.

Although I find the data interesting and valuable, I have considerable hesitation how they are attributable clinically.

6. PLOS authors have the option to publish the peer review history of their article (what does this mean?). If published, this will include your full peer review and any attached files.

Reviewer #1: No

Reviewer #2: No

Reviewer #3: No

Reviewer #4: No

---

## [Author Response · Author response to Decision Letter 0]

18 Dec 2023

Response to Reviewers

We deeply appreciate the editor and the reviewer for the critically important comments and suggestions on our paper. We have revised our manuscript according to those comments. 

All essential changes in the revised manuscript were highlighted in red font. 

Our replies to the comments and suggestions of the editor and the reviewer are written below.

Reviewer #1: The authors wrote an interesting manuscript entitled: "Variation in the frozen lesion size according to the non-occluded application duration and technique for cryoballoon ablation". This is an interesting ex vivo study that investigates the lesion size in myocardium using a cryogenic balloon with different settings.

Overall, the study appears technically sound, but does have significant limitations that the authors also addressed; these are: 1) Epicardial application of cryoenergy in ventricular myocardium. 2) Application in a non-beating model without any respiratory motion.

The scientific questions that the authors ask are all related to cryoablation in atrial myocardium, it is unclear to me why no atrial myocardium was used in this model. This would also have allowed to conduct this study from the endocardium and at the locations that are of interest for each technique; i.e. posterior wall, roof, PVs. I would encourage the authors to perform this study actually from the endocardium of atrial myocardium. They should provide us with comparison data regarding the biophysics of cryoballoon ablation in the latter.

Thank you for your important and critical comments. As you indicated, this study assessed only frozen lesion size in endocardium of ventricular myocardium, which is different from the clinical settings. However, atrial myocardium is too thin to measure the lesion depth, and endocardium was too rough to precisely measure the lesion size. Therefore, we used the endocardium of ventricular myocardium. This is the critical limitations of this study. We added the following sentences in the limitation section.

<Limitations>

First, this experimental study was conducted on the epicardial side of isolated porcine ventricular tissue, so there was no effect of the respiratory movement or cardiac beating. On the other hand, in daily clinical practice, the cryoballoon is used on the endocardial side of the atrial wall, which is thinner and softer than the ventricular wall, and with non-occluded application for atrial roof area. The frozen target and situation such as stability and direction are largely different from the non-occluded cryoballoon application in daily clinical practice use. Values such as the freezing duration and/or balloon temperature cannot be directly extrapolated to clinical use on beating human atrium. (Line 14-22 in Page 9)

I am not sure that the push up technique can be simulated in an ex vivo model because there are so many variables that impact the balloon orientation, most importantly probably position and orientation of the sheath. Therefore, these data are not easily translatable to the clinical arena.

As you indicated, the detail of the push-up technique such as direction and force is precisely defined in this study. However, we consider this vulnerable definition is to be a characteristic of the push-up technique, which can be easily performed and efficiently increase the lesion size. Therefore, through this study, we aim to demonstrate the effect of the push-up technique on lesion size. As you indicated, we added the following sentences in the Limitation section.

<Limitations>

Fourth, contact force of the cryoballoon in this study was not controlled. The difference of contact force during the freezing in various settings may influence the study results. (Line 3-5 in Page 10)

The p value <0.05 should be used as only cut-off.

As you indicated, we changed the definition of statistically significance from <0.10 to <0.05. We revised the following sentences in the Methods and Results section.

<Methods>

Probability values of p <0.05 was considered statistically significant. (Line 5-6 in Page 5)

<Methods>

However, the lesion size was not significantly different between the absence and the presence of laminar flow. (Line 13-15 in Page 9)

I would also have the language in the introduction and discussion professionally reviewed.

Our manuscript was reviewed and revised by a native English speaker, again.

Reviewer #2: Kawaji et al report on cryoballoon ablation lesion size for non-occlusive (i.e. non-PV isolation) using an ex vivo model. The authors characterize the lesion size at various application durations, using different balloon positioning techniques, and with and without laminar flow. I have the following comments:

Cryoballoon ablation is clinically used in a beating heart to treat atrial arrhythmias. This study uses an ex vivo model with ventricular tissue. The fundamental limitation of this study is the model and lack of prior validation. Cryoablation is performed in a complex anatomic and thermal environment and it is unclear whether findings from this model have clinical relevance. Furthermore, as mentioned by the authors, cryoablation is performed to chronically interrupt electrical conduction in the heart and is not certain to what degree acute lesion size, as assessed with this investigation, correlates with the intended electrical effects of ablation.

As you indicated, this study assessed only the frozen lesion size, and not the histological or electrical isolation lesion size. However, we considered the impact of various technique (application duration, push-up technique, and rapid ventricular pacing) on lesion size is same regardless of the visual and those lesion sizes. But, this is the critical limitations of this study. We revised the following sentences in the limitation section.

<Limitations>

Second, we measured only the visually frozen lesions including reversible lesion, but not the histopathologically nor electrically degenerated lesion size, which is generally considered to be smaller than the frozen lesion size. We need further study to assess the histopathological lesion size created by non-occluded cryoballoon applications. However, the sizes of those lesions are correlated with each other. The impact of the technique on frozen lesion size in this study may be consistent on these lesion sizes. (Line 22 in Page 9 – Line 1 in Page 10)

It is a bit difficult to contextualize the volume / lesion size findings in this study. When performing ablation in the atrium, the goal is for transmural lesions and a complete line of block. Lesions that are too deep/large run the risk of collateral damage to extracardiac structures/conduction system or inadvertent electrical isolation of the LAA.

As you indicated, the lesion size is large than expected. The frozen lesion size includes reversible lesion and is generally considered to be larger than the electrically isolation lesion size. In daily clinical practice use, the ice formation reached to epicardium was observed during cryoballoon application (Heart Rhythm. 2010;Oct 7(10):1518). Therefore, we consider the large frozen size is acceptable. 

There is frequent reference to the "overdrive ventricular pacing technique" but is not clear how this is any relevance to the current model, which is performed in non-beating tissue.

As you know, rapid ventricular pacing is a technique to reduce the intracardiac blood flow during catheter procedures such as TAVI, LA angiography, and roof area ablation by a cryoballoon (Nishimura et al. J Interv Card Electrophysiol 2020;59:565–573). In this study using non-beating heart, we used the laminar flow assuming the blood flow. Therefore, the situation of the absence of the laminar flow was regarded as non-circular blood flow such as rapid ventricular pacing. But, as you indicated, the absence of the laminar flow in this study may not adequately aimulate the rapid ventricular pacing in a beating heart. We added the following sentences in the Limitation section.

<Limitations>

Third, the absence of the laminar flow in this study may not adequately simulate the rapid ventricular pacing on the beating heart in daily clinical practice. (Line 1-2 in Page 10)

In the Discussion, the authors state "The push-up technique, especially at 20 seconds after freezing, was useful for creating a complete block line with nonoccluded cryoballoon applications". Unless I am missing something, there was no assessment for electrical block in this investigation; such a statement over interprets the findings.

Thank you for your important comment. The phrase is confusing expression. We revised the sentence as follows in the discussion section.

<Discussion>

The push-up technique, especially at 20 seconds after the freezing, is beneficial in creating larger frozen lesion, attributing complete block of roof line by non-occluded cryoballoon applications in daily clinical practice use (Line 17-19 in Page 8)

Reviewer #3: Interesting article that has the potential to contribute greatly to human cases of cryoablation. The cryoablation technique is relatively new and needs to be improved, and basic research helps immensely with this.

Thank you for your kind comment. As you suggested, more basic research, such as histopathological study, is required. We added the following sentences in the Limitation section.

<Limitations>

Several limitations of this experimental study should be noted. First, this experimental study was conducted on the epicardial side of isolated porcine ventricular tissue, so there was no effect of the respiratory movement or cardiac beating. On the other hand, in daily clinical practice, the cryoballoon is used on the endocardial side of the atrial wall, which is thinner and softer than the ventricular wall, and with non-occluded application for atrial roof area. The frozen target and situation such as stability and direction are largely different from the non-occluded cryoballoon application in daily clinical practice use. Values such as the freezing duration and/or balloon temperature cannot be directly extrapolated to clinical use on beating human atrium. Second, we measured only the visually frozen lesions including reversible lesion, but not the histopathologically nor electrically degenerated lesion size, which is generally considered to be smaller than the frozen lesion size. We need further study to assess the histopathological lesion size created by non-occluded cryoballoon applications. However, the sizes of those lesions are correlated with each other. The impact of the technique on frozen lesion size in this study may be consistent on these lesion sizes. Third, the absence of the laminar flow in this study may not adequately simulate the rapid ventricular pacing on the beating heart in daily clinical practice. Finally, the relatively small sample size in this study precluded drawing any definitive conclusion.

(Line 17-19 in Page 8)

Reviewer #4: The authors are attempting to create an experimental model of non-occlusive cryoablation technique using a cryoballoon and define optimal “in-vitro” ablation parameters.

It appears that optimal application duration of lesions is 150 sec with apposition against the tissue.

Methodology

Authors are attempting to approximate the in-vivo ablation environment.

Quite appropriately authors acknowledge limitations of this environment, I have several additional concerns:

1. Ventricular myocardium certainly creates a different “cold sink” than atrial myocardium.

Thank you for your important and critical comments. As you indicated, this study assessed only frozen lesion size in ventricular myocardium, which is different from the clinical settings. However, atrial myocardium is too thin to measure the lesion depth. Therefore, we used the endocardium of ventricular myocardium. This is the critical limitations of this study. We added the following sentences in the limitation section.

<Limitations>

First, this experimental study was conducted on the epicardial side of isolated porcine ventricular tissue, so there was no effect of the respiratory movement or cardiac beating. On the other hand, in daily clinical practice, the cryoballoon is used on the endocardial side of the atrial wall, which is thinner and softer than the ventricular wall, and with non-occluded application for atrial roof area. The frozen target and situation such as stability and direction are largely different from the non-occluded cryoballoon application in daily clinical practice use. Values such as the freezing duration and/or balloon temperature cannot be directly extrapolated to clinical use on beating human atrium. (Line 14-22 in Page 9)

2. Lack of tissue perfusion has potential for affecting lesion formation.

As you indicated, the different tissue perfusion between human beating heart and isolated porcine heart in this study. It is also the critical limitation. However, we consider the impact of the technique on frozen lesion size in this study may be consistent on lesion sizes in any situations.

3. There is no quantification of force applied hence potential for error.

As you indicated, the detail of the push-up technique such as direction and force is precisely defined in this study. However, we consider this vulnerable definition is to be a characteristic of the push-up technique, which can be easily performed and efficiently increase the lesion size. Therefore, through this study, we aim to demonstrate the effect of the push-up technique on lesion size. As you indicated, we added the following sentences in the Limitation section.

<Limitations>

Fourth, contact force of the cryoballoon in this study was not controlled. The difference of contact force during the freezing in various settings may influence the study results. (Line 3-5 in Page 10)

Discussion

I am surprised by the findings of no considerable difference in lesion size with or without laminar flow of the medium surrounding the balloon – authors seem not to have a good sense of what the mechanism may be. I would postulate as well to remove any implication in the discussion that these lesions are larger since they are not statistically different. Therefore, the findings here are not consistent with clinical findings quoted in reference 18.

As you mentioned, it is surprising that the absence of laminar flow did not increase lesion size. We consider that the temperature sensor is paced on the proximal side of the balloon, causing the recorded balloon temperature to predominantly reflect blood temperature rather than the actual tissue temperature. Actually, this discrepancy between freezing effectiveness and lesion size was also evident in the assessment of the impact of application duration and the push-up technique. In the clinical trial reported by Nishimura et al., rapid ventricular pacing did not affect the acute success rates in patients who received a 180-second cryoballoon application, but increased the acute completion rates in patients who experienced early interruption of the cryoballoon application due to prominent decline of esophageal temperature. Considering these insights, despite the absence of laminar flow resulting in non-significant increase in lesion depth, a decrease of blood temperature around the balloon by effective freezing may indirectly enhance freezing efficiency during non-occluded application. However, the absence of the laminar flow may not adequately simulate the rapid ventricular pacing in daily clinical practice use. We added the following sentences in the Discussion, and Limitation section.

<Discussion>

In this study, despite achieving effective freezing, the absence of laminar flow, assuming rapid ventricular pacing during non-occluded cryoballoon application, did not notably increase lesion size, except for a non-significant increase in lesion depth. This discrepancy between freezing efficacy and lesion size primarily stemmed from the placement of the temperature sensor on the proximal side of the balloon, causing the recorded balloon temperature to predominantly reflect blood temperature rather than the actual tissue temperature. This discrepancy between freezing effectiveness and lesion size was also evident in the assessment of the impact of application duration and the push-up technique. In clinical practice use, Nishimura et al. performed a randomized clinical trial to assess the utility of rapid ventricular pacing on acute success rate in achieving a complete block of the left atrial roof line by cryoballoon applications.18 Although the use of rapid ventricular pacing notably decreased the nadir balloon temperature, it did not affect the acute success rates in patients who received a 180-second cryoballoon application, but increased the acute completion rates in patients who experienced early interruption of the cryoballoon application due to prominent decline of esophageal temperature. Considering these insights, despite the absence of laminar flow resulting in non-significant increase in lesion depth, a decrease of blood temperature around the balloon by effective freezing may indirectly enhance freezing efficiency during non-occluded application.(Line 20 in Page 8 – Line 11 in Page 9)

<Limitations>

Third, the absence of the laminar flow in this study may not adequately simulate the rapid ventricular pacing on the beating heart in daily clinical practice. (Line 1-3 in Page 10)

Although I find the data interesting and valuable, I have considerable hesitation how they are attributable clinically.

Thank you for your kind comments. As you indicated, this study have several limitations, which can not be resolved. However, we consider this technique can only be experimentally validated in this study. Furthermore, it is a clinically necessary technique for left atrial roof area ablation by a cryoballoon. We added the following sentences in the Limitation section.

<Limitations>

Second, we measured only the visually frozen lesions including reversible lesion, but not the histopathologically nor electrically degenerated lesion size, which is generally considered to be smaller than the frozen lesion size. We need further study to assess the histopathological lesion size created by non-occluded cryoballoon applications. However, the sizes of those lesions are correlated with each other. The impact of the technique on frozen lesion size in this study may be consistent on these lesion sizes. (Line 22 in Page 9 – Line 1 in Page 10)

---

## [Decision Letter · Decision Letter 1]

3 Jan 2024

Variation in the frozen lesion size according to the non-occluded application duration and technique for cryoballoon ablation

PONE-D-23-36076R1

Dear Dr. Kawaji,

We’re pleased to inform you that your manuscript has been judged scientifically suitable for publication and will be formally accepted for publication once it meets all outstanding technical requirements.

Kind regards,

Gaetano Santulli, MD, PhD, FAHA

Academic Editor

PLOS ONE

Reviewers' comments:

Reviewer's Responses to Questions

**Comments to the Author**

1. If the authors have adequately addressed your comments raised in a previous round of review and you feel that this manuscript is now acceptable for publication, you may indicate that here to bypass the “Comments to the Author” section, enter your conflict of interest statement in the “Confidential to Editor” section, and submit your "Accept" recommendation.

Reviewer #4: All comments have been addressed

2. Is the manuscript technically sound, and do the data support the conclusions?

Reviewer #4: Yes

3. Has the statistical analysis been performed appropriately and rigorously? 

Reviewer #4: Yes

4. Have the authors made all data underlying the findings in their manuscript fully available?

Reviewer #4: Yes

5. Is the manuscript presented in an intelligible fashion and written in standard English?

Reviewer #4: Yes

6. Review Comments to the Author

Reviewer #4: Overall, there is an improvement in the article, certainly my concerns were addressed and other reviewers. Limitations still are in place

7. PLOS authors have the option to publish the peer review history of their article (what does this mean?). If published, this will include your full peer review and any attached files.

Reviewer #4: No

---

## [Editor Report · Acceptance letter]

17 Jan 2024

PONE-D-23-36076R1 

PLOS ONE

Dear Dr. Kawaji, 

I'm pleased to inform you that your manuscript has been deemed suitable for publication in PLOS ONE. Congratulations! Your manuscript is now being handed over to our production team.

Kind regards, 

on behalf of

Professor Gaetano Santulli 

Academic Editor

PLOS ONE